# An Overview of Instruments to Assess Vulnerability in Healthcare: A Scoping Review

**DOI:** 10.3390/healthcare13172251

**Published:** 2025-09-08

**Authors:** Filipa Andrade, Ana Resende, Clara Roquette Viana, Amélia Simões Figueiredo, Fernanda Loureiro

**Affiliations:** Centro de Investigação Interdisciplinar em Saúde (CIIS), Faculdade de Ciências da Saúde e Enfermagem (FCSE), Universidade Católica Portuguesa (UCP), 1649-023 Lisbon, Portugal; fandrade@ucp.pt (F.A.); aresende@ucp.pt (A.R.); roquetteviana@ucp.pt (C.R.V.); simoesfigueiredo@ucp.pt (A.S.F.)

**Keywords:** vulnerability, family health, health evaluation, social determinants of health, scoping review

## Abstract

**Background/Objectives**: Vulnerability is a multifaceted concept frequently addressed in healthcare, reflecting individuals’ and families’ conditions that may affect health outcomes. The availability of validated instruments to assess vulnerability is essential for supporting healthcare professionals in delivering tailored care. This scoping review aimed to map the available scientific evidence regarding vulnerability assessment instruments in individuals and/or families in the context of healthcare. **Methods**: We conducted a scoping review following the Joanna Briggs Institute methodology. A comprehensive search was carried out in the databases PubMed, SciELO, CINAHL, Scopus, and Web of Science, as well as in sources of gray literature, using relevant keywords. **Results**: A total of 38 studies were included, identifying 13 distinct instruments used to assess vulnerability at the individual and/or family level. These instruments varied widely in terms of dimensions, number of items, target populations, and modes of completion. Some instruments focused on specific aspects such as socioeconomic status, health behaviors, or access to services. **Conclusions**: The results demonstrate the complexity of the concept of vulnerability and the need to create instruments adapted to specific determinants/factors, such as environmental, biological, and social factors, as well as the specificities of target populations and contexts of assessment and intervention.

## 1. Introduction

The world’s vulnerable population has increased significantly due to shifts in global politics, geography, climate, and the effects of war, leaving many people vulnerable [1]. Vulnerability is a broad and multifaceted concept. The Oxford English Dictionary [2] defines it as the quality of being susceptible to harm, influence, or attack, as well as referring to an entity that is vulnerable. It is a concept that can be applied across various levels, such as the environment, infrastructure, communities, and individuals [3], and to several contexts such as social, economic, environmental, and healthcare [4]. This paper focuses specifically on the vulnerability of individuals and/or their families within the context of healthcare.

The concept of vulnerability is often discussed alongside other related terms such as risk, disadvantage, frailty, and marginalization. These concepts are frequently used interchangeably, but they reflect different dimensions of human experience and should be understood in their own right. Risk, for instance, refers to the potential for an event or situation to occur that may negatively affect an individual’s health, safety, or well-being [5]. Disadvantage, on the other hand, involves a condition of multiple, interconnected deprivations that undermine a person’s ability to lead a good life [6]. Frailty, typically associated with older adults, is described by the World Health Organization [7] as a state in which the ‘capacity to manage every day or acute stressors is diminished due to age-related physiological decline’. Meanwhile, marginalization refers to the social processes that exclude individuals or groups from full participation in political and social life, denying them rights, opportunities, and resources commonly available to others [8]. While these concepts intersect with vulnerability and often coexist in lived experiences, they are not synonymous. In this manuscript, our focus remains on individuals and/or their family’s vulnerability as a distinct and complex condition shaped by various personal, social, and structural factors.

In nursing, the concept of vulnerability has long been acknowledged, as ‘nursing is faced with the challenge of addressing the needs of the vulnerable’ [9]. Over time, this concept has evolved, ultimately gaining transformative power [10]. Nursing, in particular, has shown a sustained interest in this concept due to its universality, as all individuals are potentially at risk and will experience vulnerability to a greater or lesser extent [9]. Aday’s seminal work [11] highlighted several populations considered particularly vulnerable, including high-risk mothers and infants, the chronically ill and disabled, individuals with acquired immunodeficiency syndrome (AIDS), those with mental illnesses, alcohol or substance abusers, individuals prone to suicide or homicide, abusive families, the homeless, and immigrants and refugees. In our review we considered the concept of vulnerability as defined by Havrilla [9]. Based on research performed on this concept over time, this author proposes to define vulnerability as ‘a state of dynamic openness and opportunity for individuals, groups, communities, or populations to respond to community and individual factors through the use of internal and external resources in a positive (resilient) or negative (risk) manner along a continuum of illness (oppression) to health (growth)’ [9].

Vulnerability in the healthcare context is often associated with a perceived loss of power relative to others, coupled with heightened exposure to a range of risks, including physical, psychological, emotional, cultural, and economic threats [9,12,13]. The sense of vulnerability is further intensified by diminished self-confidence and challenges in navigating unfamiliar health-related situations [9,12,13]. To provide effective nursing care for vulnerable individuals, it is critical to first assess vulnerability in a comprehensive manner, ensuring that care is tailored to the unique needs of each individual. However, the measurement of vulnerability in healthcare presents significant complexities.

As previously noted, the concept of vulnerability has been explored in the literature [9,12,13,14,15]. However, the literature on methods for assessing vulnerability remains relatively sparse. In a report on vulnerability assessment methods prepared by Moret [3] for the United States Agency for International Development, the author emphasizes that vulnerability assessment at the individual level is achieved through participatory and qualitative approaches. The author also notes that the literature consistently advocates the use of mixed methods in vulnerability assessments. Levasseur and colleagues [16] assessed the literature to provide an inventory and synthesis of definitions and instruments to measure vulnerability in older adults. The authors found only five different instruments (the Hebrew Rehabilitation Center for Aged (HRCA) Vulnerability Index; the Vulnerable Elders Survey; the Social Vulnerability Index (SVI); the Perceived Vulnerability Scale (PVS); and the Vulnerability Scale for the Elderly (VSE)) and highlighted the need for measuring instruments to specify the underlying conceptual definition of vulnerability and the theoretical model following.

The use of instruments in assessment is a valuable approach given the possibility of interdisciplinary comparability [4]. By charting the available instruments, we intend to provide a valuable resource for researchers working in this field. Our aim is to map the available scientific evidence regarding vulnerability assessment instruments in individuals and/or families in the context of healthcare. Specifically, we seek to identify instruments that can be applied in future research involving vulnerable populations, whether in research studies or community-based projects.

Prior to conducting this review, we accessed public registration platforms to determine whether any ongoing or completed literature reviews were registered. These platforms included the Open Science Framework, PROSPERO, Figshare, Inplasy, the Cochrane Library of Systematic Reviews, and the Joanna Briggs Institute Systematic Review Register. No ongoing or completed reviews addressing research instruments for assessing vulnerability at the individual level and/or families in healthcare settings were identified. Therefore, a knowledge gap exists in this topic, which the authors seek to address with this review.

## 2. Materials and Methods

A scoping review [17] was conducted to identify vulnerability assessment instruments according to the Joanna Briggs Institute (JBI) methodology. The purpose of this review is to map the available scientific evidence regarding vulnerability assessment instruments in individuals and/or families in the context of healthcare. This type of review was chosen given its usefulness in the increasing of evidence synthesis approaches [18]. The research question was defined according to the PCC (P—Population; C—Concept: C—Context) mnemonic: What instruments are used to assess vulnerability in healthcare at the individual and/or family level? (Population: individuals and/or family; Concept: instruments to assess vulnerability; Context: healthcare). The steps of the scoping review, which are further explained in the checklist and explanatory paper [19], are detailed below.

### 2.1. Protocol and Registration

The scoping review protocol was developed following the Preferred Reporting Items for Scoping Reviews (PRISMA-ScR) [19] guidelines and registered with the Open Science Framework (https://osf.io/5zyem/) accessed on 21 April 2025. A supplemental file is provided to demonstrate our adherence to these guidelines (Appendix A).

### 2.2. Eligibility Criteria

We considered published articles that focused on assessing vulnerability through various instruments. Specifically, we examined the concept of vulnerability within the healthcare context, including studies where vulnerability was assessed regarding healthcare or where healthcare was one of the dimensions of vulnerability. Additionally, we intentionally employed the broad term “instruments” to encompass related terms such as scales, questionnaires, and tools. To broaden the scope of existing evidence, we included published empirical studies that employed quantitative or mixed methods. Peer-reviewed articles with no restrictions regarding publication language, type of access, or publication date were considered. Exclusion criteria included articles employing qualitative methodologies, along with letters to the editor, editorials, blog posts, advertisements, conference abstracts, materials from other types of dissemination meetings, reviews of the literature, and opinion pieces.

### 2.3. Information Sources

As recommended by Peters et al. [17], three steps were followed. In the first step, an initial search was performed in the Medical Literature Analysis and Retrieval System Online (MEDLINE) and the Cumulative Index to Nursing and Allied Health Literature (CINAHL). This search helped identify relevant keywords for the search strategy, which were then validated using Medical Subject Headings (MeSH).

In step two the research was performed in PubMed; ScieLo (via Web of Science); CINAHL [complete]; SCOPUS; Web of Science Core Collection; APA PsycArticles; and Psychology and Behavioral Sciences Collection. For gray literature OpenAIRE; ProQuest ™ Dissertations & Theses Citation Index (via Web of Science); and the Portuguese Scientific Open Access Repository (RCAAP) were consulted. In the third step, the reference lists of the selected articles were reviewed to identify additional instruments.

### 2.4. Search

Keywords such as “vulnerable”, “social vulnerability”, “vulnerable populations”, “questionnaire”, “survey”, “instrument”, “scale”, “healthcare”, “health care”, “delivery of health care”, along with boolean operators (AND/OR) and truncation using an asterisk (*) to capture word variations, were used in the search strategy. Various groupings and combinations of these terms were applied based on the specific characteristics of each platform and database. The search was conducted by FL and FA in April 2025 with the guidance of an experienced librarian. Full search strategy is fully available at Open Science Framework (https://osf.io/5zyem/, accessed on 21 April 2025).

### 2.5. Selection of Sources of Evidence

The list of references retrieved during the “search” step was imported into the Rayyan^®^ platform, where duplicates were removed. Subsequently, two researchers applied the inclusion and exclusion criteria using a double-blinded approach. Initially, articles were screened based on their titles. When it was unclear whether an article aligned with the scope of this review, the abstract was examined. To ensure consistency in the selection of sources of evidence, both authors independently reviewed the same set of publications. Any discrepancies that emerged were addressed through a structured, discussion-based approach. Instead of relying on statistical measures of agreement, we prioritized building consensus through collaborative dialog. This approach allowed us to reach a shared understanding and ensure that our decisions aligned with the review’s aim and criteria.

### 2.6. Data-Charting Process

A data-charting table was created to extract the relevant variables. This process was first carried out individually by each author, and then the results were compared to address discrepancies and improve accuracy.

### 2.7. Data Items

Data from each article was extracted related to its characteristics, namely title, year, authors, country, purpose, methods (type, population, and sample), instrument, context, results, and main conclusions. Because our aim was specifically to identify instruments, we included a second table that presents each instrument along with its key characteristics: instrument name, target population, number of items, assessment scale, reliability, validity, method of completion, and context of application.

### 2.8. Synthesis of Results

After screening each article, the results were compiled into a table that included the evidence extracted separately by each author and subsequently approved by all.

## 3. Results

### 3.1. Selection of Sources of Evidence

We conducted a comprehensive search across eight databases, and from our initial sample of 4370 articles, a total of 38 articles were included in this review. When full texts were not accessible, we contacted the corresponding authors to request access. Initially, no language restrictions were applied. However, two articles (one in Italian and one in Japanese) were subsequently excluded, as none of the research team were fluent in these languages. We also excluded studies that addressed vulnerability exclusively in relation to specific conditions or diseases, such as vulnerability to HIV. These studies tended to define vulnerability narrowly, focusing on risk factors or susceptibility related to a particular health outcome, rather than adopting a broader conceptual or multidimensional understanding of vulnerability. Additionally, we excluded articles in which vulnerability was measured using indices derived from national surveys or pre-existing questionnaires, where the authors determined which dimensions or items to include. In these cases, the assessment of vulnerability relied heavily on specific questions included in national surveys, which were tailored to the primary objectives of those surveys—objectives that did not specifically focus on the assessment of vulnerability. As a result, vulnerability was often treated as a secondary outcome, inferred from selected items or dimensions. In other instances, the assessment was based on dimensions or items chosen at the discretion of the researchers. Consequently, these approaches were not aligned with the objective of this review and could not therefore be used or adapted reliably by other researchers. We chose not to include literature reviews as a source of evidence. This decision was supported by a preliminary search conducted prior to the formal review process, which identified only one relevant literature review [16]. The same review was found again during the database and citation searches, which confirmed the lack of additional reviews addressing our research focus. As this review had already been considered and cited in the introduction, it was excluded from the screening process. Consequently, our review exclusively focused on original studies. To illustrate this process, we used PRISMA flow chart [20] as shown in Figure 1.

### 3.2. Characteristics of Sources of Evidence

A total of 13 instruments were identified and organized according to the life cycle, beginning with those designed to assess vulnerability in children, followed by instruments for adolescents, adults, and older adults. We also identified instruments specifically designed to assess vulnerability in homeless populations and in families. Main characteristics of each instrument are summarized in Table 1.

For assessment of vulnerability in children, we identified three distinct instruments: the Child Status Index (CSI) [21], the Vulnerable Child Scale [22], and the Child Vulnerability Scale (CVS) [23,24,25,26,27,28,29,30,55]. The most widely used instrument in our sample is the CVS, developed by Forsyth et al. [26], which also appears in the literature as the Forsyth Child Vulnerability Scale [55]. The scale initially comprised 12 items and aimed to assess parents’ perceptions of their child’s vulnerability. Following an assessment of its psychometric properties by the original authors, the scale was reduced to eight items. It uses a 4-point Likert scale, with total scores ranging from 0 to 24. A score of 10 or more indicates that the child is perceived as vulnerable. The scale was initially developed in the United States of America (USA) and has also been used in England [24], Norway [25], and in the Netherlands [29]. Both the CVS and the Vulnerable Child Scale are completed by the child’s parent, carer, or legal guardian. The CSI [23,24,25,26,27,28,29,30,55], however, is designed to be completed by an adult through direct questioning of either the child or the caregiver. It is also the only instrument among these three that organizes its items into distinct categories, enabling a more structured assessment.

All other instruments identified in this review were designed for use with adult populations. One exception is the study by Sass et al. [31], conducted in France, which evaluated social vulnerability among victims of interpersonal violence and considered the application of the EPICES instrument (Évaluation de la Précarité et des Inégalités de Santé dans les Centres d’Examens de Santé) to participants aged 16 years and older. EPICES consists of 11 items, with each “yes” response assigned a specific numerical weight, while “no” responses are scored as zero. The total score is calculated by summing the weights of the “yes” responses and adding a fixed constant of 75. The final score ranges from 0 to 100, with scores of 30 or higher indicating a situation of social vulnerability.

Boehm et al. [32] developed an instrument to assess USA college students’ perceptions of vulnerability or susceptibility. The instrument consists of 18 dichotomous items, with a possible total score ranging from 18 to 36. Higher scores indicate a lower perceived level of vulnerability. Tang and colleagues [33], in Canada, developed a new instrument titled the Social Vulnerabilities Survey, designed for use with adult populations. Although the authors did not specify an exact age range for respondents, the mean age of participants was 55 years. The instrument combines items adapted from previously validated questionnaires with newly developed questions.

Nineteen studies in our sample assessed vulnerability in older adults. Within this age group, we identified four distinct instruments, all of which had previously been referenced by Levasseur et al. [16], as noted in the introduction. While these authors also included the Hebrew Rehabilitation Center for the Aged Vulnerability Index, we chose to exclude this instrument from our review. Although it is labeled as an instrument for assessing vulnerability, its items primarily focus on functional status and levels of dependence, specifically evaluating aspects such as mobility and the ability to perform personal care tasks. As such, it aligns more closely with assessments of physical functionality than with broader conceptualizations of vulnerability.

The Clinical-Functional Vulnerability Index was developed in Brazil, and we identified three studies that utilized this instrument, all of which were also conducted in Brazil [35,36,37]. Each section of the instrument contributes to a total score, with a maximum possible score of 40 points. Higher scores indicate a greater risk of clinical–functional vulnerability in older adults. Individuals scoring between 7 and 14 are considered pre-vulnerable, while those with a score of 15 or higher are classified as vulnerable. The Perceived Vulnerability Scale (PVS) was developed in Australia [34] and consists of 22 self-administered items. Responses are rated on a 6-point Likert scale, ranging from 1 (not at all vulnerable) to 6 (extremely vulnerable). Higher total scores indicate a greater perceived vulnerability. Kaushik [38] proposed an instrument based on a theoretical framework for item development; however, the instrument was not tested or validated.

Among the instruments targeting older adults, the most frequently used in the studies we retrieved was the Vulnerable Elders Scale (VES) [39,40,41,42,43,44,45,46,47,48,49,50,51]. This instrument, originally developed by Saliba et al. [45], is commonly referred to as VES-13 due to its 13 items, which are distributed across four dimensions. Each item contributes a specific number of points based on the response, and the total score ranges from 0 (no vulnerability) to 10 (high vulnerability). A score of 3 or higher is considered indicative of vulnerability.

Two researchers from the USA, Bowie & Lawson [52], utilized a pre-existing instrument known as the Vulnerability Index (VI), originally developed to assess and prioritize individuals in need of housing based on their risk of mortality. In their study, the authors explored the feasibility of applying the VI to assess the healthcare needs of a local homeless community. They concluded that the instrument was easy to administer and effective in generating a health needs profile for this population.

To assess family vulnerability, we located two instruments: the Family Vulnerability Scale (FVS) [53,56,57] and the Index of Family Development [54,58,59]. Both instruments were developed in Brazil. The FVS includes 14 items, each scored as either 0 or 1. The total score classifies families into three levels of vulnerability: low vulnerability (0–4), moderate vulnerability (5–6), and high vulnerability (7–14). The Index of Family Development initially comprised eight domains and 103 items [54]. Following further refinement, the instrument was reduced to 50 items distributed across seven factors [58]. In the most recent publication by the same author [59], cut-off points were established and validated for each dimension. A family was classified as vulnerable if it scored 15 or more points on the total scale. Additionally, a family was considered vulnerable in health conditions if it scored six or more points within that specific dimension.

### 3.3. Results of Individual Sources of Evidence

For each article retrieved, we extracted relevant data, which is summarized in a table publicly available through our OSF registration. Additionally, given that our objective was to map the available instruments used to assess vulnerability, we compiled Table 1, which presents the main characteristics of each identified instrument.

### 3.4. Synthesis of Results

We retrieved studies published between 1993 [32] and 2025 [47] from a variety of countries. The countries with the highest number of studies were the USA (18 studies) [22,23,26,27,28,30,32,41,42,43,44,45,46,47,48,50,52,55] and Brazil (9 studies) [35,36,37,53,54,56,57,58,59]. All retrieved studies were quantitative in nature, with the exception of the article by Kaushik [38], which was theoretical. Most studies (25) assessed vulnerability in a community context [21,23,24,25,32,34,35,36,37,38,39,40,41,42,43,44,45,52,53,54,55,56,57,58,59], while only 14 were conducted in or applicable to hospital settings [22,26,27,28,29,30,31,33,46,47,48,49,50,51].

## 4. Discussion

This scoping review allowed us to map a diverse set of instruments used to assess the vulnerability of individuals and/or families in the context of healthcare. The data analysis revealed significant heterogeneity among the identified instruments. This variability was evident in several aspects, including the presence or absence of specific instruments, the definitions of the dimensions assessed (i.e., categories of biological and social determinants of health influencing vulnerability), and their structural composition (such as the number of items per dimension covering relevant topics within each category). Differences were also observed in the types of assessment scales used, the reliability and validity of the instruments, as well as in their method of application and the context in which they were developed. This diversity, far from representing a limitation, reflects the complexity of the concept of vulnerability and the need for approaches adapted to the specificities of each population and reality [10,11,12,15]. It is also important to recognize that as a broad, multifaceted concept, vulnerability is influenced by numerous factors including, but not limited to, gender identity, migration status, and type of disability. This makes assessing vulnerability an inherently contextual task.

Analysis of the dimensions covered by the instruments revealed considerable variability in how vulnerability assessment was operationalized as their primary objective. Some have defined dimensions, while others do not clearly report the dimensions that comprise them. When reported, these reflect, in some way, but not systematically, some of the health/social determinants that influence the health status of the target populations, such as the following: biological determinants (age) and social determinants (living conditions: housing, nutrition, transportation; socioeconomic conditions: education/vocational training, remuneration, social support; health systems: access to protective care/physical and psychosocial health; cultural context: access to cultural habits, self-perception/self-assessment of health; social factors: social network support). Others also assess very specific aspects of health conditions (functional impairments, cognitive impairments, mood, mobility, communication, aging, chronic diseases, and comorbidities). This diversity can be interpreted as a reflection of the attempt to adapt the instruments to specific contexts and populations, but it also raises questions about the comparability of data and the possibility of developing a universal instrument. The variation in the number of dimensions, between four and eight, directly influences the depth of the assessment, requiring a balance between comprehensiveness and feasibility [14,38,58].

Another relevant aspect concerns the number of items in the instruments, which vary between eight and fifty questions/topics. More extensive instruments may offer a more comprehensive assessment, but they may be less realistic in their applicability in contexts with time or resource constraints, especially when applied to vulnerable populations [16]. On the other hand, while shorter instruments may facilitate ease of application, they may fail to capture all relevant dimensions of vulnerability. The lack of reported data on average administration time in most studies represents a significant limitation, as it impedes the evaluation of these instruments’ suitability. Therefore, it is recommended that future research systematically report this information to better inform the suitability of these instruments for use in practice. Advances in digital health technologies, such as computer-adaptive testing and machine learning-based scoring systems, offer promising alternatives to the traditional trade-off between instrument length and precision. These technologies enable brief assessments to achieve high measurement accuracy.

The modes of instrument completion also varied considerably, with some designed for self-administration by participants, while others were completed by health professionals, caregivers, or researchers. Self-administration can enhance autonomy and capture the individual’s subjective perception of vulnerability; however, it is also susceptible to biases, such as socially desirable responses or difficulties in comprehending the instrument’s dimensions and items. Conversely, administration by third parties, such as caregivers, researchers, or health professionals, may introduce subjective interpretations shaped by their own beliefs and experiences, particularly when they lack experience working with vulnerable populations. The types of response scales employed (dichotomous, ordinal, and Likert) each present distinct advantages and limitations. Dichotomous scales offer simplicity and ease of use but tend to lack sensitivity. In contrast, ordinal and Likert scales provide greater sensitivity and nuance in data collection, though they demand a higher level of interpretive skill from respondents and require greater complexity in analysis.

The use of varied response scales in data collection provides several advantages, enabling researchers to obtain richer and more nuanced information, minimize bias, and enhance the reliability of findings. The selection of an appropriate scale should be guided by the research objectives, the nature of the data to be collected, and the characteristics of the target population. Careful consideration of each scale’s validity and reliability is essential to ensure the quality and interpretability of the results [60,61].

This review identified only a few instruments with defined cut-off scores linked to specific intervention pathways: the Family Vulnerability Scale [35,53] and the Index of Family Development [54]. This type of linkage improves the practical application of assessment instruments and is considered best practice. The future development and refinement of instruments should prioritize establishing empirically supported criteria to guide clinical and policy decisions, thereby making vulnerability assessments more actionable and relevant in real-world settings.

The applicability of vulnerability assessment instruments is heavily influenced by the sociocultural context in which they were developed. The predominance of instruments originating from the USA raises concerns about their relevance and suitability in other settings, particularly in European contexts. As vulnerability is shaped by health determinants such as biology, society, and the environment, effective assessment instruments must reflect the specific priorities and characteristics of the target population and context. Directly transposing these tools to different cultural or population contexts may compromise their validity, highlighting the need for careful cultural adaptation and rigorous validation processes. The limited availability of instruments developed or validated within European settings highlights a significant gap in the field, emphasizing the importance of investing in context-specific research and instrument development [12]. We also have to consider other contexts with vulnerable populations such as low- and middle-income countries. In these contexts, key aspects such as translation, cultural relevance, and contextual sensitivity, aligned with current best practices (e.g., cognitive debriefing and differential item functioning testing), are essential components of future research in this area. We also underscore the need for more instrument development and validation work originating from these countries to ensure global applicability and equity in vulnerability assessment.

In this context, it becomes evident that the development of a single, universally accepted instrument for assessing vulnerability is both conceptually reductive and methodologically impractical. Vulnerability is a multidimensional phenomenon that manifests differently across populations and settings, shaped by a complex interplay of determinants. These include biological factors such as age, sex, genetic predisposition, and exposure to health risks; social determinants including the social environment (e.g., housing conditions, sanitation, access to essential services such as clean water), physical environment (e.g., availability of green spaces, transportation infrastructure, food access), socioeconomic status (e.g., education level, employment status, income, and social security coverage), healthcare system (e.g., access to quality care, type of health coverage), lifestyle behaviors (e.g., diet, physical activity, alcohol and tobacco use, sleep patterns, leisure activities), cultural context (e.g., religious beliefs, cultural practices, values, and social norms), social support networks (e.g., formal and informal support, relationship dynamics, experiences of discrimination or social exclusion), and public policies (e.g., health, education, housing policies); and environmental/physical determinants such as geographical and climatic conditions (e.g., altitude, climate variability, hydrographic systems, and soil characteristics) [62,63,64].

Given this diversity, any attempt to standardize vulnerability assessment into a single instrument risk overlooking crucial context-specific factors. Instead, tailored approaches that reflect the unique characteristics and priorities of different populations and settings are essential for valid and meaningful assessment [9].

The assessment of the psychometric properties of the instruments generally revealed satisfactory levels of validity and reliability. Most studies report on content validity, construct validity, and internal consistency, which confers methodological robustness on the analyzed instruments. However, not all instruments provide a complete description of their psychometric properties, and few studies address the cross-cultural validity of the determinants of vulnerability outside the USA. This limits the applicability of the instruments outside their original context, reinforcing the need for rigorous validation and cultural adaptation in future research.

Another interesting aspect to highlight in this review is that most studies assessed vulnerability in the context of community healthcare. According to the literature, vulnerable populations, particularly homeless populations, stand out in this context, particularly in southern European countries and some African and Asian countries, and to a lesser extent in hospital or inpatient settings [65,66,67]. We have to consider the importance of cultural and policy context in interpreting and applying vulnerability assessment instruments. Future validation efforts should include careful examination of potential item bias across different geographic and sociopolitical settings. Although the date of instrument development does not limit its selection, it is also an aspect that researchers must consider, as older tools may not accurately reflect current realities, changes in healthcare systems, social conditions, or population needs.

Beyond mapping what exists we propose a preliminary decision-making framework that considers several key factors to guide the selection of the most appropriate tool:Target population: Table 1 presents the instruments organized by population type (e.g., children, older adults), which can help professionals quickly identify instruments suited to specific demographic groups.Depth of assessment required: Considering the number of items and the dimensions assessed can help researchers choose between brief screening and more comprehensive evaluations, depending on the goals of the assessment and the context in which it is applied.Available institutional resources: Some instruments are self-administered, while others require trained researchers or professionals to administer them. This may involve additional resource allocation and associated costs, which should be considered during planning.Mode of administration (digital vs. paper-based): While most instruments reviewed did not specify the mode of administration, this is an important consideration, especially in contexts involving vulnerable populations who may face barriers to digital access (e.g., limited internet access, low digital literacy, or language constraints).

Selecting a validated instrument that is well-suited to the specific context can help identify vulnerable situations and support the implementation of more effective, targeted interventions. When cultural adaptation is necessary, researchers should follow established guidelines for translation and cross-cultural adaptation. They should also pay particular attention to the context-specific nature of the concept of vulnerability. As our review has shown, the dimensions that define vulnerability can vary significantly across cultural, social, and institutional settings. This makes contextual sensitivity a critical factor in ensuring that the adapted instrument remains valid and meaningful for the target population.

We also recognize the importance of ongoing access to up-to-date information and advocate for the future development of a living, open access database that compiles psychometric properties, target populations, languages, and context of use for each instrument identified in this review. Such a platform would serve as a dynamic resource for researchers and practitioners, supporting evidence-based selection and adaptation of vulnerability assessment instruments in diverse healthcare settings.

From a research perspective, future studies could build upon the findings of this review by conducting an appraisal of measurement properties or by applying methods such as the Jaccard index to quantify conceptual overlap among instruments. Such approaches would support the development of more precise, targeted, and efficient assessment tools.

This review highlights the importance of exercising caution when selecting existing instruments and when designing new ones to assess vulnerability in individuals and families across the life course within healthcare contexts. New instruments should be grounded in clearly defined and structured dimensions, aligned with key categories of health and social determinants—biological, environmental (including climatic and physical), and social—relevant to specific regional contexts. Additionally, they should include an appropriate number of items sensitive to the specific themes within each dimension.

These tools should be reliable, capable of validly assessing the same factors/determinants that influence vulnerability, and replicable in different geographical and population contexts, with the ability to select the most appropriate dimensions and questions according to these differences. Furthermore, future efforts should aim to validate existing instruments in new contexts to expand their applicability and ensure cultural and contextual relevance.

To synthesize our main findings in this review, we employed the framework proposed by Bradbury-Jones et al. [68] that identifies patterns, advances, gaps, evidence for practice, and research recommendations (PAGER), as outlined in Table 2.

This scoping review was conducted with methodological rigor. However, certain limitations must be acknowledged. Relevant studies may have been unintentionally excluded due to limitations in the scoping review protocol. Despite developing a comprehensive search strategy with an experienced librarian and using multiple databases, additional instruments might have been identified with broader databases or more diverse language inclusion. Some studies failed to clearly describe the characteristics of the instruments used, such as completion time or required reading grade level, or to provide access to the instruments themselves. This posed challenges in extracting and analyzing key information. This lack of transparency limited our ability to fully assess and compare instruments. We were also unable to retrieve a small number of potentially relevant studies, as several authors did not respond to our repeated contact attempts. Despite multiple follow-up efforts, this lack of correspondence further restricted access to complete data.

## 5. Conclusions

This scoping review enabled us to identify and critically analyze the most recent instruments used to assess vulnerability in various contexts and populations. The results demonstrate the complexity of the concept of vulnerability and the need to create instruments adapted to specific determinants/factors, such as environmental, biological and social factors, as well as the specificities of target populations and contexts of assessment and intervention.

The absence of a universal instrument and the variability in the dimensions, items, and scales used reinforce the importance of a careful and contextualized choice of instruments. This is particularly important given the gaps in the cross-cultural validation of the impact of different determinants/factors that influence vulnerability in each population and context of assessment and intervention.

This review not only contributes to clinical practice by providing an updated survey of the available vulnerability assessment instruments, but also points the way to future research, particularly in the development of culturally sensitive and methodologically robust instruments. In this sense, we propose the development of an instrument to assess vulnerability throughout the life cycle of individuals and families in the context of healthcare, with a particular focus on the home and community. This approach is consistent with the majority of the results of this review.

Within the scope of the dimensions, the construction of this instrument will take into account the classification of health determinants, namely biological and social determinants, which influence the expression of vulnerability. The items will align with the vulnerability assessment in each dimension and be appropriate for the target population (children and families), with an appropriate scale for community-based assessment and intervention and the available resources. Professionals with experience in community intervention and with vulnerable populations should carry out the vulnerability assessment instrument for individuals and/or families throughout their life cycle in the context of healthcare.

## Figures and Tables

**Figure 1 healthcare-13-02251-f001:**
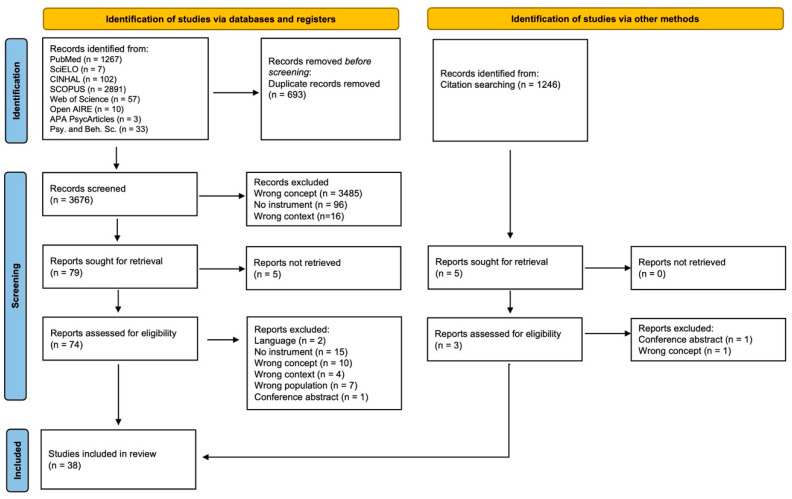
PRISMA flow chart for study selection.

**Table 1 healthcare-13-02251-t001:** Characteristics of retrieved instruments used to assess vulnerability.

Instrument Name	Population	Dimensions	Items	Assessment Scale	Reliability	Validity	Distribution/Completion	Context of Application
Child Status Index	Children	(1) Food/nutrition (2) Shelter and care (3) Protection (4) Health (5) Psychosocial situation (6) Education and skills training	12	4-point Likert scale	Not reported	Construct validity	Filled by adults by asking questions directly to the children or to the caregiver	Community [21]
Vulnerable Child Scale	Children	Not reported	15	5-point Likert scale	Not reported	Not reported	Filled by parents or legal guardians of children	Hospital [22]
Child Vulnerability Scale	Children	Not reported	8	4-point Likert scale	Internal consistency	Content validity Criterion-related validity Construct validity Concurrent Face validity	Filled by parents	Community [23,24,25] Hospital [26,27,28,29,30]
Évaluation de la Précarité et des Inégalités de Santé dans les Centres d’Examens de Santé	Adults or teenagers ≥ 16 ages	(1) Material situation (2) Social situation (3) Education (4) Access to culture	11	Dichotomous (Yes/No)	Internal consistency	Content validity Construct validity Concurrent validity Face validity	Filled by participants	Hospital [31]
Vulnerability/Susceptibility Scale	Adults 18–22 years	(1) Basic Needs (2) Major Stress (3) Minor Illness (4) Diet (5) Minor Stress	18	Dichotomous (Yes/No)	Internal consistency	Construct validity	Filled by participants	Community [32]
Social Vulnerabilities Survey	Adults	(1) Salience or priority of health (2) Social support (3) Transportation (4) Finances	33	Categorical, ordinal, and continuous	Internal consistency	Face validity Construct validity	Filled by participants	Hospital [33]
Perceived Vulnerability Scale	Adults ≥ 50 years	Not reported	22	6-point Likert scale	Internal consistency and test–retest	Construct validity	Filled by participants	Community [34]
Clinical-Functional Vulnerability Index	Adults ≥ 60 years	(1) Age (2) Health self-perception (3) Functional disabilities (4) Cognition (5) Mood (6) Mobility (7) Communication (8) Comorbidities	20	Each section has a specific score	Internal consistency	Not reported	Applied by a health professional/researcher	Community [35,36,37]
Vulnerability scale for the elderly	Elderly	(1) Health vulnerability (2) Social vulnerability (3) Economic vulnerability (4) Overall vulnerability	12	Categorical, ordinal, and continuous	Not reported	Not reported	Not reported	Not reported [38]
Vulnerable Elders Scale	Adults ≥ 60 years	(1) Age (2) Self-rated health (3) Limitation in physical function (4) Functional disabilities	13	Cumulative and ordinal	Internal consistency	Construct validity	Filled by participants or by a health professional	Community [39,40,41,42,43,44,45] Hospital [46,47,48,49,50,51]
Vulnerability Index	Homeless ≥ 18 years	(1) Demographic information (2) Housing history (3) Health questions (4) Medical conditions (5) Social history	34	Yes/No/Refused	Not reported	Not reported	Filled by the individual or by a researcher/professional	Community [52]
Family Vulnerability Scale	Families	(1) Income (2) Healthcare (3) Family (4) Violence	14	Dichotomous (Yes/No)	Internal consistency	Content validity Construct validity	Filled by a family member	Community [35,53]
Index of Family Development	Families	(1) Favorable social conditions (2) Aging (3) Chronic diseases (4) Unfavorable social conditions (5) Social support (6) Illiteracy (7) Social network	50	Dichotomous (Yes/No)	Internal consistency	Content validity Construct validity Concurrent validity	Filled by a family member	Community [54]

**Table 2 healthcare-13-02251-t002:** Summary of the review according to the PAGER framework.

Patterns	Advances	Gaps	Evidence for Practice	Research Recommendations
Diversity in instrument structure and dimensions.	The development of instruments adapted to different populations and contexts.	There is a lack of consensus on essential dimensions and a lack of standardization.	The choice of instrument should consider the target population and specific context.	Investigate which dimensions are most relevant in different cultural and clinical contexts.
Number of items and feasibility.	The existence of short and long instruments allows for some flexibility.	There is a lack of standardization of appropriate items by dimension. There is a lack of data on application time. Longer instruments may be less feasible.	Shorter instruments may be more appropriate in contexts with different resource constraints.	Include application time systematically in validation studies.
Methods of completion and types of scales.	Use of different scales (e.g., dichotomous, ordinal, Likert) and completion methods.	The possibility of bias depends on who completes the form and the lack of standardization.	The choice of scale and completion method should consider the literacy level and profile of the population.	Study the impact of the completion method on the validity of the data collected.
Cultural and contextual applicability.	Instruments developed in contexts with high population diversity (e.g., the USA).	There is a lack of validation in European contexts, which poses a risk of cultural inadequacy.	Instruments must be culturally adapted before they can be applied in other contexts.	Develop and validate instruments in European and multicultural contexts.
Psychometric properties and validation.	Many instruments demonstrate content validity and internal consistency.	Not all studies clearly describe their validation methods, and there is a lack of cross-cultural validation.	Using validated instruments increases the reliability of data.	Rigorous validation and cross-cultural adaptations of existing instruments should be promoted.

## Data Availability

No new data were created or analyzed in this study. Data sharing is not applicable to this article.

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
