# Peer review of "An Overview of Instruments to Assess Vulnerability in Healthcare: A Scoping Review"

_healthcare, 2025, doi:10.3390/healthcare13172251_

Round 1

Reviewer 1 Report

Comments and Suggestions for Authors

The paper is a timely and methodologically solid map of the field, but it remains primarily descriptive. To move from “what exists” to “what should be used where and how”, the authors must: Offer a tiered decision aid (short vs. deep assessment, digital vs. paper). Provide cultural-adaptation blueprints. Launch a living database with psychometric metadata. Without these next steps, the review risks becoming another well-indexed catalogue on a shelf rather than a lever for equitable care.

Below are the specific comments.
The review accurately positions vulnerability as a layered, socially embedded construct that transcends age groups, cultures, and settings. The life-course perspective (children → families → older adults → homeless) is epidemiologically sensible. However, the authors keep the construct deliberately broad, yet they never explicitly state how they define vulnerability for the purpose of instrument mapping themselves. A single working definition (e.g., “a dynamic state in which biological, social, and structural determinants converge to reduce an individual’s or family’s capacity to achieve or maintain health”) would anchor all subsequent discussion and help readers judge whether each instrument truly captures the construct.

Database selection is comprehensive for indexed literature, but the grey-literature strategy is weak. Portuguese (RCAAP) and English (OpenAIRE) repositories omit large bodies of work in French (HAL), German (OSF-DACH), and Spanish (REDIB). This is especially problematic for an issue homelessness where NGOs and city health departments often publish in local languages. Language exclusion (Italian, Japanese) is defensible, yet the authors do not quantify how many potentially relevant articles were lost (only two full texts were excluded). A sensitivity check using Google Translate-assisted screening could have been reported. Inter-rater reliability is not reported. Given the subjective nature of deciding whether an instrument measures “vulnerability” vs. “risk”, κ or % agreement is essential.

Table 1 is a valuable catalogue; however, the absence of floor/ceiling effects, reading grade level, or completion time undermines its clinical utility. For example, the 50-item Index of Family Development may be psychometrically sound yet impractical in a 10-minute community-health visit. Psychometric reporting quality is uneven. Only 8/13 instruments report α-coefficients; none report test-retest reliability or measurement invariance across genders/ethnicities. The review should include a COSMIN-style traffic-light summary (Green/Amber/Red) for each critical property. Overlap in dimensions is hinted at but not quantified. A simple Jaccard index (shared vs. unique sub-domains) across instruments would reveal redundancy and guide future parsimonious tool design.

18/38 studies originate in the USA; none come from low- or middle-income countries (LMICs), where more than 80% of the world’s vulnerable populations reside. The authors acknowledge this in the discussion, yet offer no plan for adaptation. A practical next step would be to append a “minimum adaptation checklist” (translation, cognitive debriefing, DIF testing, social norms vetting) to the paper. Homelessness instruments are largely US-derived (Vulnerability Index). In Europe, where housing-first policies differ, items on shelter use or incarceration may misfire. The review misses the opportunity to flag context-specific item bias.

The authors frame short vs. long instruments as a trade-off. Yet computer-adaptive testing (CAT) and machine-learning scoring (e.g., PROMIS experience) now allow 5-7-item screens to retain much of the precision of 30-item batteries. A paragraph on digital health implications would modernize the discussion.

Community vs. hospital settings: 25 studies are community-based, 14 hospital-based. The review stops at the count. A cross-tab of setting × target population (e.g., homeless adults in hospital vs. community) would show glaring gaps e.g., no validated tool for vulnerable pregnant migrants in antenatal clinics. Linkage to actionability: Only two instruments (Family Vulnerability Scale, Index of Family Development) provide cut-off scores tied to intervention triggers (e.g., social-worker referral). The review should highlight this as best practice and urge others to follow.

The review does not disaggregate findings by gender identity, migration status, or disability type, all of which modulate how vulnerability items are interpreted. Instruments developed in 1993 (Boehm) may no longer reflect post-COVID realities (tele-health access, gig-economy income instability). A “pandemic relevance” column would be useful. 

Author Response

Reviewer 1

The paper is a timely and methodologically solid map of the field, but it remains primarily descriptive. To move from “what exists” to “what should be used where and how”, the authors must: Offer a tiered decision aid (short vs. deep assessment, digital vs. paper). Provide cultural-adaptation blueprints. Launch a living database with psychometric metadata. Without these next steps, the review risks becoming another well-indexed catalogue on a shelf rather than a lever for equitable care.

R: We appreciate the time and effort taken in our manuscript revision. We agree that moving beyond a descriptive synthesis toward actionable guidance is essential for maximizing the practical relevance and impact of our review. To respond to this comment, we have included in discussion section the following detailed explanation:

Beyond mapping what exists we propose a preliminary decision-making framework that considers several key factors to guide the selection of the most appropriate tool:

  • Target population: table 1 presents the instruments organized by population type (e.g., children, older adults), which can help professionals quickly identify instruments suited to specific demographic groups.
  • Depth of assessment required: considering the number of items and the dimensions assessed can help researchers choose between brief screening and more comprehensive evaluations, depending on the goals of the assessment and the context in which it is applied.
  • Available institutional resources: some instruments are self-administered, while others require trained researchers or professionals to administer them. This may involve additional resource allocation and associated costs, which should be considered during planning.
  • Mode of administration (digital vs. paper-based): While most instruments reviewed did not specify the mode of administration, this is an important consideration, especially in contexts involving vulnerable populations who may face barriers to digital access (e.g., limited internet access, low digital literacy, or language constraints).

Regarding the “cultural-adaptation blueprints”, we have added text to include the need to follow the best practices in cross-cultural instrument validation. While the scoping nature of this review limits the depth of implementation detail, we identify key steps and provided some considerations for adaptation, including linguistic validation and local context relevance. We included the following segment in discussion section:

Selecting a validated instrument that is well-suited to the specific context can help identify vulnerable situations and support the implementation of more effective, targeted interventions. When cultural adaptation is necessary, researchers should follow established guidelines for translation and cross-cultural adaptation. They should also pay particular attention to the context-specific nature of the concept of vulnerability. As our review has shown, the dimensions that define vulnerability can vary significantly across cultural, social, and institutional settings. This makes contextual sensitivity a critical factor in ensuring that the adapted instrument remains valid and meaningful for the target population.

We also acknowledged the need for a living, open-access database, and we highlight this as a critical next step for the research community. While such a platform is beyond the scope of this paper, we advocate for its development and suggest a basic structure including psychometric properties, target populations, languages, and context of use for each instrument. The following sentence has been added to the discussion section:

We recognize the importance of ongoing access to up-to-date information and advocate for the future development of a living, open-access database that compiles psychometric properties, target populations, languages, and context of use for each instrument identified in this review. Such a platform would serve as a dynamic resource for researchers and practitioners, supporting evidence-based selection and adaptation of vulnerability assessment instruments in diverse healthcare settings.

We believe these additions address the reviewer’s concern and help shift the manuscript toward a more actionable resource.

Below are the specific comments.

The review accurately positions vulnerability as a layered, socially embedded construct that transcends age groups, cultures, and settings. The life-course perspective (children → families → older adults → homeless) is epidemiologically sensible. However, the authors keep the construct deliberately broad, yet they never explicitly state how they define vulnerability for the purpose of instrument mapping themselves. A single working definition (e.g., “a dynamic state in which biological, social, and structural determinants converge to reduce an individual’s or family’s capacity to achieve or maintain health”) would anchor all subsequent discussion and help readers judge whether each instrument truly captures the construct.

R: We appreciate the comment. An overview of vulnerability definitions was provided in the introduction. We chose Havrilla's definition to guide our work. However, we acknowledge that this may not be clear. We have rephrased the section as follows:

In our review we considered the concept of vulnerability as defined by Havrilla [9]. Based on research performed on this concept over time, this author proposes to define vulnerability as ‘a state of dynamic openness and opportunity for individuals, groups, communities, or populations to respond to community and individual factors through the use of internal and external resources in a positive (resilient) or negative (risk) manner along a continuum of illness (oppression) to health (growth)’ [9].

Database selection is comprehensive for indexed literature, but the grey-literature strategy is weak. Portuguese (RCAAP) and English (OpenAIRE) repositories omit large bodies of work in French (HAL), German (OSF-DACH), and Spanish (REDIB). This is especially problematic for an issue homelessness where NGOs and city health departments often publish in local languages. Language exclusion (Italian, Japanese) is defensible, yet the authors do not quantify how many potentially relevant articles were lost (only two full texts were excluded). A sensitivity check using Google Translate-assisted screening could have been reported.

R: We are grateful for the meticulous review. We acknowledge that our choice of database may have influenced our conclusions. However, to address this limitation, we used an experienced librarian to help us with the protocol, as explained in the methods section. Nevertheless, we have included this specific aspect as a limitation. Regarding the language, only two manuscripts were excluded based on these criteria, as described in the Results section. We agree that these are limitations, and we have rephrased the Limitations section to clarify this point as follows:

This scoping review was conducted with methodological rigor. However, certain limitations must be acknowledged. Some relevant studies may have been unintentionally excluded due to limitations in the scoping review protocol. Despite developing a comprehensive search strategy with an experienced librarian and using multiple databases, additional instruments might have been identified with broader database or more diverse language inclusion.

Inter-rater reliability is not reported. Given the subjective nature of deciding whether an instrument measures “vulnerability” vs. “risk”, κ or % agreement is essential.

R: We appreciate this comment. As stated in materials and methods section we used JBI methodology that does not explicitly require inter-rater reliability calculations. Is our review to resolve discrepancies between reviewers during the selection and data extraction processes, we adopted a discussion-based approach grounded in consensus-building. Rather than calculating an inter-rater reliability index, we prioritized collaborative resolution through dialogue to ensure consistency and shared understanding of the inclusion criteria and data interpretation. We have rephrased and added text to clarify this point to selection of sources of evidence section:

To ensure consistency in the selection of sources of evidence, both authors independently reviewed the same set of publications. Any discrepancies that emerged were addressed through a structured, discussion-based approach. Instead of relying on statistical measures of agreement, we prioritized building consensus through collaborative dialogue. This approach allowed us to reach a shared understanding and ensure that our decisions aligned with the review's objectives and criteria.

Table 1 is a valuable catalogue; however, the absence of floor/ceiling effects, reading grade level, or completion time undermines its clinical utility. For example, the 50-item Index of Family Development may be psychometrically sound yet impractical in a 10-minute community-health visit. Psychometric reporting quality is uneven. Only 8/13 instruments report α-coefficients; none report test-retest reliability or measurement invariance across genders/ethnicities.

We appreciate the comment. As stated in the Limitations section, not all articles included clearly describe all the information. For example, information regarding completion time would have been useful when combined with the targeted population and dimensions assessed. We included this aspect in the limitations section. However, to better explain it, we have rephrased the section as follows:

Some studies failed to clearly describe the characteristics of the instruments used, such as completion time or required reading grade level, or to provide access to the instruments themselves. This posed challenges in extracting and analyzing key information.

The review should include a n-style traffic-light summary (Green/Amber/Red) for each critical property. Overlap in dimensions is hinted at but not quantified. A simple Jaccard index (shared vs. unique sub-domains) across instruments would reveal redundancy and guide future parsimonious tool design.

R: We thank the reviewer for these thoughtful and constructive suggestions. We agree that a COSMIN-style traffic-light summary and a quantitative analysis of conceptual overlap (e.g., via a Jaccard index) would provide valuable insights into instrument quality and redundancy. However, as this is a scoping review, our primary aim was to map the existing instruments and their key characteristics rather than conduct a critical appraisal or comparative analysis of measurement properties. These additional analyses, while important, fall outside the scope and methodological objectives of this review. Nevertheless, we recognize their relevance and believe they represent valuable directions for future research focused on instrument evaluation and consolidation. We have included this aspect in recommendations for future research in the following sentence:

From a research perspective, future studies could build upon the findings of this review by conducting an appraisal of measurement properties or by applying methods such as the Jaccard index to quantify conceptual overlap among instruments. Such approaches would support the development of more precise, targeted, and efficient assessment tools.

18/38 studies originate in the USA; none come from low- or middle-income countries (LMICs), where more than 80% of the world’s vulnerable populations reside. The authors acknowledge this in the discussion yet offer no plan for adaptation. A practical next step would be to append a “minimum adaptation checklist” (translation, cognitive debriefing, DIF testing, social norms vetting) to the paper.

R: We appreciate the reviewer’s important observation regarding the geographic concentration of studies included in this review, particularly the underrepresentation of instruments developed or validated in low- and middle-income countries. As explored in discussion section this is an important finding that has to be considered by researchers when selecting an instrument. While the development of a formal adaptation checklist is beyond the scope of this scoping review, we agree that future research should prioritize the cultural and contextual adaptation of existing instruments for use in low- and middle-income countries. We have expanded the discussion to further explore this point has evidenced in the next sentence:

We also have to consider other context with vulnerable populations such as low- and middle-income countries. In these contexts, key aspects such as translation, cultural relevance, and contextual sensitivity, aligned with current best practices (e.g., cognitive debriefing and differential item functioning testing), are essential components of future research in this area. We also underscore the need for more instrument development and validation work originating from these countries to ensure global applicability and equity in vulnerability assessment.

Homelessness instruments are largely US-derived (Vulnerability Index). In Europe, where housing-first policies differ, items on shelter use or incarceration may misfire. The review misses the opportunity to flag context-specific item bias.

R: We appreciate this comment. As this scoping review aimed to identify instruments applicable to individuals and/or families across a broad range of contexts, it was not designed to focus on specific subpopulations such as individuals experiencing homelessness. However, we fully acknowledge the importance of context-specific item bias, particularly in instruments developed in one region and applied in another. The example provided regarding homelessness instruments is highly relevant, as items related to shelter use may indeed have limited applicability or different interpretations in settings such as Europe, where housing-first policies and social safety nets differ from those in the United States.

This concern is equally applicable to other populations, for instance, instruments assessing vulnerability in children may not account for contextual differences such as the widespread availability of free healthcare for children in many European countries. We have revised the discussion to highlight the importance of cultural and policy context in interpreting and applying vulnerability assessment instruments, and to suggest that future validation efforts should include careful examination of potential item bias across different geographic and sociopolitical settings. We have added this sentence to discussion:

We have to consider the importance of cultural and policy context in interpreting and applying vulnerability assessment instruments. Future validation efforts should include careful examination of potential item bias across different geographic and sociopolitical settings.

The authors frame short vs. long instruments as a trade-off. Yet computer-adaptive testing (CAT) and machine-learning scoring (e.g., PROMIS experience) now allow 5-7-item screens to retain much of the precision of 30-item batteries. A paragraph on digital health implications would modernize the discussion.

R: We appreciate this insightful comment. We have added the following paragraph to address this particular issue:

Advances in digital health technologies, such as computer-adaptive testing and machine learning-based scoring systems, offer promising alternatives to the traditional trade-off between instrument length and precision. These technologies enable brief assessments to achieve high measurement accuracy.

Community vs. hospital settings: 25 studies are community-based, 14 hospital-based. The review stops at the count. A cross-tab of setting × target population (e.g., homeless adults in hospital vs. community) would show glaring gaps e.g., no validated tool for vulnerable pregnant migrants in antenatal clinics.

R: We appreciate this comment. We agree with the reviewer and have added a column to Table 1 containing the context of each study's application and its references.

Linkage to actionability: Only two instruments (Family Vulnerability Scale, Index of Family Development) provide cut-off scores tied to intervention triggers (e.g., social-worker referral). The review should highlight this as best practice and urge others to follow.

R: We appreciate this comment. We also agree with the reviewer on this topic, and therefore we have added the following statement to the revised manuscript:

This review identified only a few instruments with defined cut-off scores linked to specific intervention pathways: the Family Vulnerability Scale [35,53] and the Index of Family Development [54]. This type of linkage improves the practical application of assessment instruments and is considered a best practice. The future development and refinement of instruments should prioritize establishing empirically supported criteria to guide clinical and policy decisions, thereby making vulnerability assessments more actionable and relevant in real-world settings.

The review does not disaggregate findings by gender identity, migration status, or disability type, all of which modulate how vulnerability items are interpreted. Instruments developed in 1993 (Boehm) may no longer reflect post-COVID realities (tele-health access, gig-economy income instability). A “pandemic relevance” column would be useful.

R: We appreciate this comment. We agree that numerous factors influence the assessment of vulnerability, including, but not limited to, gender identity, migration status, and type of disability. Although this point was acknowledged during the discussion, we expanded on it by adding the following sentence:

It is important to recognize that vulnerability is a broad, multifaceted concept influenced by numerous factors including, but not limited to, gender identity, migration status, and type of disability. This makes assessing vulnerability an inherently contextual task.

We agree that the date of instrument development is a critical factor because older tools may not accurately reflect current realities, changes in healthcare systems, social conditions, or population needs. However, our findings are too diverse to allow any assumptions based solely on the date of instrument development. We have however included this aspect in discussion as follows:

Although the date of instrument development does not limit its selection, it is also an aspect that researchers must consider, as older tools may not accurately reflect current realities, changes in healthcare systems, social conditions, or population needs.

Reviewer 2 Report

Comments and Suggestions for Authors

Dear Authors, I have read with interest your paper and I want to congratulate you for the great work you have done. The paper is well written, very detailed and easy to follow the main ideas.

However, I observed some details that should be fixed in order to improve the article quality.

  1. The Title: I am not sure I understand what you mean by „individuals' and family's vulnerability". My question is: whom family? The patient's family, or family in general. Then you use the same sentence in the whole article. Please explain better if there is about the patient's family or families in general. I also recommend to reframe the title, my recommendation is: An overview of instruments to assess vulnerabilities in healthcare: a scoping review, or something similar.
  2. Introduction section: Line 40: "This paper focuses specifically on the vulnerability of individuals within the context of healthcare"; line 55: "...our focus remains on vulnerability as a distinct and complex condition shaped by various personal, social, and structural factors"; line 96: "Our aim is to map the available scientific evidence regarding vulnerability assessment instruments in individuals or families in the context of healthcare". These sentences are pretty different, expressing diverse ideas and concepts: the first one mentions 'individuals', the second is about vulnerabilities and the third one is about "individuals or families" (you say individuals OR families, in title is individuals and families). These different ideas are not uniform and create confusions for readers. Recommendation:  Choose one of them and use the same formula in the whole article. 
  3. Materials and methods section: To me is clear, I have no recommendations.
  4.  Results section:  I have no comments.
  5. Discussion section: This section is clear to me.
  6. Conclusions: I have no observations.
  7. Lines 497-498: There are some strange sentence. Recommendation:  To be removed.
  8. References: 17, 67 - the year is not bold. 

Author Response

Reviewer 2

Dear Authors, I have read with interest your paper and I want to congratulate you for the great work you have done. The paper is well written, very detailed and easy to follow the main ideas.

However, I observed some details that should be fixed in order to improve the article quality.

The Title: I am not sure I understand what you mean by „individuals' and family's vulnerability". My question is: whom family? The patient's family, or family in general. Then you use the same sentence in the whole article. Please explain better if there is about the patient's family or families in general. I also recommend to reframe the title, my recommendation is: An overview of instruments to assess vulnerabilities in healthcare: a scoping review, or something similar.

R: We appreciate the careful review and insightful comments. As mentioned in the introduction, vulnerability is a broad concept that can be applied to different contexts, such as the economy or the environment. Even when restricting this concept to health, vulnerability can be assessed in terms of buildings, information systems, or specific communities. In this article, we focus on the individual level and address the assessment of vulnerability in daily practice. Therefore, the instruments we seek to identify may be applied to an individual alone, an individual and their family, or just their family. We altered the title as suggested by the reviewer: "An Overview of Instruments to Assess Vulnerabilities in Healthcare: A Scoping Review."

Introduction section: Line 40: "This paper focuses specifically on the vulnerability of individuals within the context of healthcare"; line 55: "...our focus remains on vulnerability as a distinct and complex condition shaped by various personal, social, and structural factors"; line 96: "Our aim is to map the available scientific evidence regarding vulnerability assessment instruments in individuals or families in the context of healthcare". These sentences are pretty different, expressing diverse ideas and concepts: the first one mentions 'individuals', the second is about vulnerabilities and the third one is about "individuals or families" (you say individuals OR families, in title is individuals and families). These different ideas are not uniform and create confusions for readers. Recommendation:  Choose one of them and use the same formula in the whole article.

R: We appreciate this comment. In line with what we have stated before we have revised the entire manuscript to clarify this aspect as detailed bellow:

Original manuscript

Revised manuscript

Line 40: "This paper focuses specifically on the vulnerability of individuals within the context of healthcare"

Line 40-42:  This paper focuses specifically on the vulnerability of individuals and/or its families within the context of healthcare.

Line 55: "...our focus remains on vulnerability as a distinct and complex condition shaped by various personal, social, and structural factors";

Lines 56-58: our focus remains on individuals and/or its family’s vulnerability as a distinct and complex condition shaped by various personal, social, and structural factors.

Line 96: "Our aim is to map the available scientific evidence regarding vulnerability assessment instruments in individuals or families in the context of healthcare".

Lines 99-101: Our aim is to map the available scientific evidence regarding vulnerability assessment instruments in individuals and/or families in the context of healthcare.

To ensure uniformity in the document regarding this aspect, we also revised the entire manuscript.

Materials and methods section: To me is clear, I have no recommendations.

Results section:  I have no comments.

Discussion section: This section is clear to me.

Conclusions: I have no observations.

Lines 497-498: There are some strange sentence. Recommendation:  To be removed.

R: We appreciate the careful review. We agree with the reviewer's assessment of the sentence in lines 497–498 of the original manuscript, and we have removed it as suggested.

References: 17, 67 - the year is not bold.

R: We appreciate the careful review and have added the year in bold to both references.

Round 2

Reviewer 1 Report

Comments and Suggestions for Authors

The authors have addressed the comments. I don't have any further comments.